# The Relationship between Self-Rated Economic Status and Falls among the Elderly in Shandong Province, China

**DOI:** 10.3390/ijerph17062150

**Published:** 2020-03-24

**Authors:** Zhuang Hong, Lingzhong Xu, Jinling Zhou, Long Sun, Jiajia Li, Jiao Zhang, Fangfang Hu, Zhaorong Gao

**Affiliations:** 1School of Public Health, Shandong University, Jinan 250012, China; 201835802@mail.sdu.edu.cn (Z.H.); jj_lin123@126.com (L.S.); lfyx_hzzz@163.com (J.L.); 201614324@mail.sdu.edu.cn (J.Z.); hufang1218@yeah.net (F.H.); gzr201314@mail.sdu.edu.cn (Z.G.); 2NHC, Key Laboratory of Health Economics and Policy Research, Shandong University, Jinan 250012, China; 3Center for Health Economics Experiment and Public Policy Research, Shandong University, Jinan 250012, China; 4School of Medicine and Health Management, Shandong University, Jinan 250012, China; jianggsxcwm@yeah.net

**Keywords:** elderly, falls, risk factors, rural and urban, self-rated economic status

## Abstract

(1) Background: Older people are more vulnerable and likely to have falls and the consequences of these falls place a heavy burden on individuals, families and society. Many factors directly or indirectly affect the prevalence of falls. The aims of this study were to understand the prevalence and risk factors of falls among the elderly in Shandong, China; the relationship between economic level and falls was also preliminary explored. (2) Methods: Using a multi-stage stratified sampling method, 7070 elderly people aged 60 and over were selected in Shandong Province, China. General characteristics and a self-rated economic status were collected through face to face interviews. Chi-square tests, rank sum tests and two logistic regression models were performed as the main statistical methods. (3) Results: 8.59% of participants reported that they had experienced at least one fall in the past half year. There was a significant difference in experienced falls regarding gender, residence, marital status, educational level, smoking, drinking, hypertension, diabetes, coronary disease, and self-reported hearing. The worse the self-rated economic status, the higher the risk of falling, (poor and worried about livelihood, OR = 3.60, 95%; CI = 1.76–7.35). (4) Conclusions: Women, hypertension, diabetes and self-reported hearing loss were identified as the risk factors of falls in the elderly. The difference of economic level affects the falls of the elderly in rural and urban areas. More fall prevention measures should be provided for the elderly in poverty.

## 1. Introduction

Population aging is rapidly developing in the world, especially in China. It is estimated that the percentage of people in China aged 60 years and above will increase from 12.4% (168 million) in 2010 to 28% (402 million) in 2040 [1]. This considerable increase in the elderly population will soon trigger a global burden of disease, and falls are a major issue of the global burden undergone by the elderly population [2,3,4]. Not only are older people more vulnerable to falls than younger people, but also, the elderly have more injuries due to falls and a high medical utilization rate. In 2016, unintentional falls accounted for 27.7% of all nonfatal-injury emergency department visits by adults 18 years and older, and 62.2% of all nonfatal-injury emergency department visits for adults 65 years and older [5]. Functional loss, resulting from fall events including traumatic brain injury, hip fractures, and other moderate to severe injuries, can result in nursing home placement, restricted activity, loss of autonomy, and death in the elderly population [6,7,8,9]. Falls are the most costly type of injury among the elderly, placing a heavy burden on individuals, families and society [10].

Every year, accidental falls occur in nearly one-third of the elderly, and in 50% of adults over the age of 80 [11,12]. Falls in adults are the result of a combination of direct and indirect risk factors [13,14]. In addition to biological characteristics (e.g., age and sex) interacting with modifiable physical characteristics (e.g., diabetes, depression, decline of hearing and cognitive capacities, intake of multiple medications, and drinking) [15,16], environmental factors, such as poor lighting and uneven surfaces, are all known risk factors for falls among older adults [17,18]. In addition to those, evidence suggests that the incidence of falls varies among people of different economic levels, including children and the elderly [19,20,21].

After entering the 21st century, the economy has developed rapidly in both rural and urban areas in China, but the gap between the rural and urban economies is widening [22]. For reasons, such as the difference in housing provided to urban employees, employment income and pension, the economic status of the urban elderly is better than that of the rural elderly [23]. In view of the economic inequality between the rural and urban elderly, this difference may lead to the differences in the incidence of falls. The aims of this study were therefore to investigate the incidence of falls and the associated risk factors of the elderly population in Shandong province, China; and to explore whether self-rated economic status differences affect the rural and urban elderly falls.

## 2. Methods

### 2.1. Study Design and Data Collection

This was a population-based study of subjects of 60 years of age and over in Weihai, Weifang and Heze, Shandong province, China. Fall screening data from a total of 7070 elderly persons who went through the complete face-to-face interviews were used. The areas were chosen according to the level (high, medium and low) of per capita gross domestic product (GDP) in Shandong province, and the geographical position (eastern, central and western) of Shandong province. We used a multi-stage stratified sampling method to extract a representative sample of the elderly over 60 from these three areas. Firstly, one district and one county were randomly selected from the three cities. Secondly, three streets (towns) were randomly selected in each sample area (county). Thirdly, in each sample street (township), 6 villages (residences) were randomly selected. Finally, 50 households aged 60 and above were randomly selected in each village (residence). Details of the study design and data collection methods are described elsewhere [24,25]. After preinvestigation and strict sampling, 7088 elderly people were included in the study, and 18 of them were excluded because they had not completed all of the interviews. In summary, a total of 7070 residents were eligible for the study.

The verbal informed consent was obtained from each participant prior to the survey and Academic Research Ethics Committee of Shandong University confirmed this form of consent and approved the study protocol.

### 2.2. Socio-Demographic Characteristics

In this study, a structured questionnaire was used to obtain demographic data (gender, age, residence, living arrangement, marital status, educational level, smoking behavior, drinking behavior, hypertension, diabetes, coronary disease, self-reported hearing and personal self-rated economic status). Height without shoes was measured in centimeters (accuracy 1.0 cm), and weight in light clothing was measured in kilograms (accuracy 0.01 kg). Body Mass Index (BMI) was calculated as weight divided by height squared (kg/m^2^) [24]. 

According to the official resident registration database (Hukou), the location of residence was classified into rural and urban. The living arrangements of the elderly in rural and urban areas were classified as living alone and not living alone. Marital status was dichotomized into couple and single (including unmarried, divorced, widowed or other). As for educational level, we divided it into primary school and below, junior high school, and high school and above. Behavior risk factors included smoking status (current, never, or former) and drinking status (current, never, or former). Drinking and smoking behaviors were self-reported by the respondents. We asked “do you have the habit of smoking and drinking now?” and the respondents replied “Yes or No”, and “Did you smoke before?” and respondents replied "Yes or No". Drinking behavior and smoking behavior were obtained by the same way of asking questions. Patients with diseases were also self-reported. The investigator asked “Do you have hypertension confirmed by the doctor or health professional?” and the respondents replied “Yes or No”. The same was asked for diabetes and coronary disease. We considered respondents to have had a fall experience if they reported at least one fall in the last 6 months. We defined hearing loss as self-reported trouble hearing (“How do you think of your hearing, normal, moderate trouble or a lot of trouble?”).

### 2.3. Self-Rated Economic Status

In this study, economic status was examined through one question, “What do you think of your financial situation?”. Respondents could answer “Wealthy and not worried about livelihood”, “Not wealthy but not worried about livelihood”, “Not wealthy and worried about livelihood” or “Poor and worried about livelihood”.

### 2.4. Statistical Analysis

First, descriptive statistics are presented as the number of cases, percentage, and means with standard deviations (SD). Second, the incidence of fall in subjects with different characteristics was expressed as a percentage of the study population, and association with gender, age, residence, living arrangement, marital status, educational level, smoking behavior, drinking behavior, self-reported chronic conditions, self-reported hearing and self-rated economic status was assessed by a chi-square test. A continuous variable (BMI) was compared by a rank sum test. Lastly, using logistic regression, we obtained odds ratios (ORs) for the association between self-rated economic status and falls in two adjusted models. Model 1 adjusted for significant variables from the univariate analyses; gender, residence, marital status, educational level, smoking behavior, drinking behavior, self-reported chronic conditions and self-reported hearing were included. Model 2 additionally adjusted for self-rated economic status in order to examine how much of the variance in this association was explained by economic factors. We used SPSS version 22.0 for all analyses. We calculated 95% confidence intervals (CIs) and considered two-sided *p* values < 0.05 to be statistically significant.

## 3. Results

### 3.1. Social-Demographic Characteristics of the Sample

We included 7070 elderly responders in our study sample (Table 1), all of whom are over 60 years old. In total, 8.59% reported that they had fallen in the past half year. The majority of our respondents had the following characteristics: female (59.75%), living in rural areas (77.99%), not living alone (85.49%), couple (81.17%), primary and below education (73.47%), self-reported economic level as not wealthy but not worried about livelihood (69.59%), never smoked (71.13%), never drank (75.88%), no hypertension (55.81%), no diabetes (85.83%), no coronary disease (78.32%), and normal hearing (76.69%). Of the 607 (8.59%) elderly who reported falls in the past six months, most of the social-demographic characteristics showed differences, except for age, BMI and living arrangements. There was a significant difference in experiencing falls regarding gender (<0.001), residence (<0.05), marital status (<0.001), educational level (<0.001), smoking (<0.01), drinking (<0.001), hypertension (<0.01), diabetes (<0.01), coronary disease (<0.01), self-reported hearing (<0.01), and self-rated economic status (<0.001). Relatively speaking, the elderly who are female, rural residents, in a couple, have primary and below education, never smoked, never drank, with hypertension, with no diabetes, with no coronary disease, normal hearing, and not wealthy but not worried about livelihood, were associated with a greater risk of fall.

Respondents who experienced a fall in the past six months reported a higher frequency of hearing loss (include moderate or severe hearing loss) compared to those without any falls (29.32% vs. 22.74%). In Model 1, after adjusting for gender, marital status, education level, smoking, drinking, hypertension, diabetes, coronary disease, and self-reported hearing, the relationship between moderate hearing loss and falls still exists (*p* = 0.003), but the relationship between severe hearing loss and falls is no longer significant (*p* = 0.140). Meanwhile, after adjusting for related factors, the odds of a fall experience in the past six months were 0.79 times lower among the urban elderly compared to rural (95% CI = 0.63–1.00, *p* < 0.05) (Model 1, Table 2). Results were changed after adjusting for self-rated economic status (OR = 0.91; 95% CI = 0.72–1.16; *p* = 0.462), and there was no significant difference in falls between the rural and urban elderly (Model 2, Table 2).

### 3.2. The Association between Economic Status and Falls

At the same time, in Model 2, which included adjustment for self-rated economic status, there was support for a link between moderate hearing loss and falls (OR = 1.30; 95% CI = 1.06−1.59; *p* = 0.01). Regarding economic level, those with fall experience in the past six months were significantly worse on the self-rated economic status than the counterpart (not wealthy but not worried about livelihood, OR = 1.54, 95% CI = 1.20−1.97, *p* = 0.001; not wealthy and worried about livelihood, OR = 2.40, 95% CI = 1.69−3.42, *p* < 0.001; poor and worried about livelihood, OR = 3.60, 95% CI = 1.76−7.35, *p* < 0.001).

## 4. Discussion

Falls are frequent events; however, as they are multifactorial, it is difficult to establish a single risk factor for their occurrence [26]. This study investigated the incidence of falls and associated risk factors, and explored whether self-rated economic status differences affect the rural and urban elderly falls. In a previous study on the assessment of the prevalence and factors associated with falls in 2096 elderly in various states of Nigeria, it was found that the major risk factors for falls were female gender, advanced age, and low or medium socioeconomic status [27]. Other studies have found greater propensity of falls among people with poor hearing compared to normal hearing [28].

In this study, we observed the effects of gender, age, BMI, residence, living arrangements, marital status, educational level, smoking behavior, drinking behavior, some diseases (hypertension, diabetes, coronary disease), self-reported hearing, and personal self-rated economic status on the occurrence of falls. The results showed that five risk factors (gender, hypertension, diabetes, self-reported hearing, and self-rated economic status) had a relationship with the occurrence of falls, and may provide additional opportunities for fall prevention and risk factor management. These results were consistent with a falls risk assessment study conducted in Tianjin, China, with a sample of 619 subjects over 60 years old. This study showed a higher prevalence of falls associated with gender, fall history, and diabetes [29].

In the present study, the fall rate (8.59%) was lower than those that found in other studies focusing on other countries’ elderly, such as those in Australia (30%) [30], Thailand (18.7%) [31], and China (20.1%) [29]. This difference in fall incidence may be due to the fact that the fall experiences used in this study were in the last six months, rather than in the last year, as in those previous studies. However, in terms of reducing recall bias, this study may be more credible.

On the one hand, several factors that were not found to be associated with fall events, including age, personal lifestyle (e.g., smoking history, alcohol consumption) and anthropometry (e.g., body mass index), were consistent with the findings of previous studies [32,33]. On the other hand, a cohort study demonstrates that gender was significantly associated with increased risk of accidental falls [14], where being female was more susceptible to falls (HR = 1.52). This result was consistent with the findings in the present study, even after adjusting the variables that statistically differ in univariate analysis. Similarly, in a study of the elderly 65 years and above, the women’s falls incidence exceeded men’s in each age group [34]. Other studies have shown that this difference between men and women can be explained by physiological characteristics and changes in bones, muscle structure and hormones associated with menopause [26,35,36] Among the individual (intrinsic) risk factors associated to the events of falls discussed in the literature, is chronic disease, including hypertension, osteoporosis, rheumatic diseases and diabetes [16,26,29].

In our study, the results of univariate analysis suggested that hearing loss was related to falls. In the further adjusted variable model, hearing loss (moderate) was still a risk factor for falls. A previous systematic review study found a significant association between hearing loss and falls [37]. Meanwhile, in a survey of adults by Elizabeth R. Heitz, it was suggested that hearing loss may be a clinical indicator of increased fall risk [16]. In some studies, the reasons why hearing loss affects falls were explained. Loss of hearing may also directly prevent people from cues that are needed for environmental awareness. The association of hearing loss with falls may be mediated through cognitive load and reduced attentional resources [27]. Attentional resources are key to maintaining postural control, and degradation in attentional and cognitive resources imposed by hearing loss may be harmful to the maintenance of postural balance in real-life situations and increase the risk of falling [38,39].

This study also revealed that another factor that may contribute to the risk of falls was economic status, because, compared with the affluent elderly, those in poor economic conditions lack some facilities to prevent falling, such as railings, walking aids and crutches. Meanwhile, the previous literature points out that the absence of devices like hand rails on stairs, or anti-skid surfaces in baths and showers, can increase the risk of falls [40]. In addition, it has also been pointed out in some of the literature that economic level is associated with falls [41,42,43]. People with different economic statuses may live in different environments; for example, in the quality of flooring and lighting in their residences, access to public transportation, and recreational areas, which may also contribute to the occurrence of falls in the elderly [16]. As a protective factor of falls [44,45], exercise also has differences in economic level. Poor economic conditions will affect exercises [46], thus increasing the risk of falls.

## 5. Strengths and Limitations

There are some limitations in our research. First, although in some studies personal physical and mental health are important, we did not evaluate them in this study. Further evaluation about the effects of combined health conditions and social factors to falls should be investigated. However, we adjusted for numerous confounders and used a large and representative sample, which minimizes the likelihood of these explanations for our findings. Second, our analyses are cross-sectional, and do not permit the determination of the cause–effect relationship of falls. However, considering that a fall is an acute event, and that economic status is relatively long term, we take that it is reasonable to assume that our results reflect the impact of self-rated economic status on falls, rather than the reverse. Third, because the habits of smoking and drinking are considered to be non-compliant with social norms, the respondents may be inclined to underreport these behaviors. Finally, given that the data collected from the elderly on their past fall history, there is an inevitable recall bias. While some studies have analyzed the relationship between falls and physical conditions, few have analyzed the relationship between economic conditions and falls, especially among the older Chinese, with large economic disparities in rural and urban areas. This study supports the association between hearing loss and falls and explores the effects of economic factors on falls in the elderly, providing additional targets for preventing falls in elderly people. Economic factors as a possible risk factor for falls in the elderly need to be determined by further research.

## 6. Conclusions

The results from our findings supported the conclusion that elderly women with hypertension, diabetes, and self-reported hearing loss are more susceptible to suffer falls than their counterparts. There were significant differences between self-rated economic status and falls among the elderly in Shandong province, China. Economic status may be a risk factor for falls—the poorer people are, the more likely they are to fall. The results of this study suggest that elderly fall prevention strategies such as allowance, hypertension and diabetes care, and early hearing detection and intervention need to be developed.

## Figures and Tables

**Table 1 ijerph-17-02150-t001:** Prevalence of falls in the past six months by participant characteristics among the elderly in Shandong, China (*n* = 7070).

	*n*	No (%)	Yes (%)	*χ* ^2^	*p*-Value
Total	7070	6463 (94.41)	607 (8.59)		
Gender				27.2876	0.000
Male	2846 (40.25)	2662 (41.19)	184 (30.31)		
Female	4224 (59.75)	3801 (58.81)	423 (69.69)		
Age	69.81 ± 6.45	69.75 ± 6.42	70.38 ± 6.73	2.3885	0.122
Body Mass Index (BMI)	24.69 ± 3.76	24.65 ± 3.73	25.05 ± 4.02	2.1294	0.145
Residence				6.3494	0.012
Rural	5514 (77.99)	5016(77.61)	498 (82.04)		
Urban	1556 (22.01)	1447 (22.39)	109 (17.96)		
Living alone				0.6940	0.405
No	6044 (85.49)	5532 (85.59)	512 (84.35)		
yes	1026 (14.51)	931 (14.41)	95 (15.65)		
Marital status				13.4138	0.000
Single	1331 (18.83)	1183 (18.30)	148 (24.38)		
Couple	5739 (81.17)	5280 (81.70)	459 (75.62)		
Educational level				17.9513	0.000
Primary school and below	5194 (73.47)	4704 (72.78)	490 (80.72)		
Junior high school	1315 (18.60)	1233 (19.08)	82 (13.51)		
High school and above	561 (7.93)	526 (8.14)	35 (5.77)		
Smoking				13.1582	0.001
Current	1138 (16.10)	1058 (16.37)	80 (13.18)		
Never	5029 (71.13)	4559 (70.54)	470 (77.43)		
Former	903 (12.77)	846 (13.09)	57 (9.39)		
Drinking				18.5411	0.000
Current	1153 (16.31)	1089 (16.85)	64 (10.54)		
Never	5365 (75.88)	4863 (75.24)	502 (82.70)		
Former	552 (7.81)	511 (7.91)	41 (6.75)		
Hypertension				11.5676	0.001
No	3946 (55.81)	3647 (56.43)	299 (49.26)		
Yes	3124 (44.19)	2816 (43.57)	308 (50.74)		
Diabetes				9.2395	0.002
No	6068 (85.83)	5572 (86.21)	496 (81.71)		
Yes	1002 (14.17)	891 (13.79)	111 (18.29)		
Coronary disease				11.8270	0.001
No	5537 (78.32)	5095 (78.83)	442 (72.82)		
Yes	1533 (21.68)	1368 (21.17)	165 (27.18)		
Self-reported hearing				13.4482	0.001
Normal	5422 (76.69)	4993 (77.26)	429 (70.68)		
Moderate trouble	1411 (19.96)	1259 (19.48)	152 (25.04)		
A lot of trouble	237 (3.35)	211 (3.26)	26 (4.28)		
Self-rated economic status				48.8293	0.000
Wealthy and not worried about livelihood	1618 (22.89)	1530 (23.67)	88 (14.50)		
Not wealthy but not worried about livelihood	4920 (69.59)	4479 (69.30)	441 (72.65)		
Not wealthy and worried about livelihood	478 (6.76)	411 (6.36)	67 (11.04)		
Poor and worried about livelihood	54 (0.76)	43 (0.67)	11 (1.81)		

**Table 2 ijerph-17-02150-t002:** Associations between self-rated economic status and falls in the past six months among the elderly in Shandong, China.

Variable	Model 1			Model 2		
	*β*	OR	95% CI	*β*	OR	95% CI
Gender						
Male		1.00			1.00	
Female	0.39	1.48 **	1.13−1.94	0.39	1.48 **	1.13−1.94
Residence						
Rural		1.00			1.00	
Urban	−0.24	0.79 *	0.63−1.00	−0.09	0.91	0.72−1.16
Marital status						
Single		1.00			1.00	
Couple	−0.22	0.81 *	0.66−0.99	−0.20	0.82	0.67−1.00
Educational level						
Primary school and below		1.00			1.00	
Junior high school	−0.25	0.78	0.60−1.00	−0.21	0.81	0.63−1.04
High school and above	−0.16	0.85	0.59−1.24	−0.06	0.94	0.64−1.37
Smoking						
No ^a^		1.00			1.00	
Yes ^b^	0.09	1.10	0.92−1.30	0.08	1.09	0.91−1.29
Drinking						
No^a^		1.00			1.00	
Yes^b^	−0.12	0.88	0.75−1.04	−0.11	0.90	0.76−1.06
Hypertension						
No		1.00			1.00	
Yes	−0.18	0.84 *	0.71−1.00	−0.17	0.84 *	0.70−1.00
Diabetes						
No		1.00			1.00	
Yes	−0.29	0.75 *	0.60−0.94	−0.29	0.74 **	0.59−0.93
Coronary disease						
No		1.00			1.00	
Yes	−0.22	0.80 *	0.66−0.98	−0.19	0.83	0.68−1.00
Self−reported hearing						
Normal		1.00			1.00	
Moderate trouble	0.31	1.36 **	1.11−1.66	0.26	1.30 **	1.06−1.59
A lot of trouble	0.32	1.38	0.90−2.10	0.28	1.32	0.86−2.02
Self−rated economic status						
Wealthy and not worried about livelihood					1.00	
Not wealthy but not worried about livelihood				0.43	1.54 ***	1.20−1.97
Not wealthy and worried about livelihood				0.88	2.40 ***	1.69−3.42
Poor and worried about livelihood				1.28	3.60 ***	1.76−7.35

* *p* ≤ 0.05, ** *p* ≤ 0.01 and *** *p* ≤ 0.001; ^a^ never, ^b^ former or current.

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
