# Peer review of "The Relationship between Self-Rated Economic Status and Falls among the Elderly in Shandong Province, China"

_ijerph, 2020, doi:10.3390/ijerph17062150_

Round 1
Reviewer 1 Report
I have read it carefully and with interest. The article is interesting for the journal and deals with an interesting issue. The introduction outlines well the subject and is up-to-date from a bibliographical point of view.
The theme is of foremost importance to public health. The text is well organized and reads smoothly.
The Introduction appropriately justified the relevance of the study.
Page 2 line 48: It seems to me that the verb and the conclusion of the sentence are missing
The materials and methods are well described, the analysis was well conducted and manuscript well written.
Page 10 line 32: there is a typo "Error!...."
The authors within the conclusion correctly point out what the limits of the study are for them.
In my opinion, the manuscript should be accepted after minor revision.
Reviewer 2 Report
Using a stratified sample of 7070 adults aged 60 and over from Shandong Province, China, the presence study examines determinants and risk factors for the occurrence of falls among the elderly. The independent variables include gender, age (grouped in 5 year intervals), BMI, residence, household structure (living alone vs. else), marital status (single vs. not), educational attainment, substance use, chronic morbidity (hypertension, diabetes, coronary disease), hearing ability and economic status. The results indicate that the occurrence of falls is significantly higher among women, those suffering from hypertension, diabetes and self-reported hearing loss.
This study has several methodological limitations. First of all, it is not clear to me why did the authors decided to treat age at the ordinal level. Treating age as an interval/ratio variable instead is preferable because this does not reduce variation. Was the sample originally stratified by age at 5 year intervals? Second, there is a concern for collinearity because the two variables, living arrangement and marital status, can have overlapping categories. To be exact, people who live alone are very likely to be single. Likewise, coupled individuals are very unlikely to live alone. Combining these two variables into one will make sure that correlation between living arrangement and marital status does not induce undue collinearity in the sample. Finally, the authors do not specify whether such behavioral factors as drinking and smoking are based on self-reports. This is important because, in general, respondents have the tendency to underreport their drinking and smoking habit (in order to comply with social norms). Likewise, it is not stated whether the information on hypertension was derived from self-reports. I believe if the authors can address or, at least, mention these limitations, the paper can be published.
Round 2
Reviewer 2 Report
It appears that the authors tackled a number of methodological challenges and problems, and, consequently, I believe that the revised manuscript has been improved. However, it is not clear to me what the following sentence mean:
P. 3, line 95: So do diabetes and coronary disease.
Please rewrite.